# Modes of administering sexual health and blood-borne virus surveys in migrant populations: A scoping review

Daniel Vujcich[1]*, Sonam Wangda[2], Meagan Roberts[1], Roanna Lobo[1], Bruce Maycock[3], Chanaka Kulappu Thanthirige[1], Alison Reid[1]

1 School of Public Health, Curtin University, Perth, Australia, 2 Ministry of Health, Thimphu, Bhutan, 3 College of Medicine & Health, University of Exeter, Exeter, United Kingdom

* daniel.vujcich@curtin.edu.au

**Data Availability Statement:** All relevant data are within the paper and its Supporting Information files.

## Abstract

There has been a growing number of sexual health and blood-borne virus (SHBBV) surveys specifically administered to migrant populations. The purpose of this scoping review is to collate available information about how SHBBV surveys have been administered in migrant populations and the effect that mode of administration has on data quality, reliability and other practical considerations, e.g. response rates (RR) and social desirability bias. A methodological framework for scoping reviews was applied. SHBBV survey studies administered to international migrants published since 2000 were included if they contained some description of mode of administration. Ninety one studies were identified for inclusion from Embase, Medline, Web of Science, Google Scholar and supplementary grey literature. 'Interview only' was the most common mode of administration (n = 48), predominately comprising face-to-face interviews. Thirty six studies reported data from 'self-completed' surveys only, with pen-and-paper being most common (n = 17). Few studies (n = 7) combined interview and self-completed methods of survey administration. Sixty one studies did not report (or only partially reported) RR or the data necessary to calculate RR. Of the studies that reported RR, most were missing other key information including method of recruitment, consent procedures and whether incentives were offered. Strengths and limitations of all administration modes are summarised. Guidelines to inform future SHBBV survey research in migrant populations are presented.

## Introduction

Migrants are a priority group for the prevention and control of HIV/AIDS [1]. Between 2007 and 2012, 42% of HIV diagnoses in Western Europe were in migrant populations [2]. Elsewhere such as in United States of America and Australia, migrants accounted for 19% and 38% of HIV diagnoses respectively [3, 4]. Existing research suggests that migrants may encounter legal, social, economic and cultural barriers to healthcare access in relation to HIV and other sexually transmissible infections and blood-borne viruses [5–7].

**Funding:** AR, BM, RL, DV and MR received funding from the Australia Research Council: https://www.arc.gov.au/. Additional project funding was provided by the Department of Health Western Australia (https://ww2.health.wa.gov.au/), the Department of Health South Australia (https://www.sahealth.sa.gov.au/), the Department of Health and Human Services Victoria (https://www.sahealth.sa.gov.au/) and SHine SA (https://www.shinesa.org.au/) The funders had no role in study design, data collection and analysis, decision to publish, or preparation of the manuscript.

**Competing interests:** The authors have declared that no competing interests exist.

In spite of the priority for this population, migrants are often under-represented in research, including in the context of general population sexual health and blood-borne virus (SHBBV) surveys [8–10]. High quality data are needed to monitor whether strategic objectives relating to this population group are being met or need to be adjusted in response to changing circumstances. As such, there has been a growing number of SHBBV surveys specifically developed for migrant populations, including the African Health and Sex Survey in England, the Advancing Migrant Access to Health Services in Europe (aMASE) study and the HIV community survey in people from culturally and linguistically diverse backgrounds in New South Wales, Australia [11–13]. Additionally, the World Health Organisation is in the process of developing a standard instrument for measuring sexual health knowledge, practices and outcomes worldwide, and has sought submissions on implementation factors including survey administration channels [14, 15].

While there are a range of factors which can affect the quality of survey data (e.g. validity of survey constructs, sampling and recruitment methods), the focus of this article is the mode of survey administration. As a recent literature review shows, the manner of survey administration can greatly affect the quality of the data collected by influencing response rates, completion rates, respondent cognition and social desirability bias [16]. However, this review did not seek to determine whether certain modes of administration were more appropriate for specific topic areas, especially those of a sensitive nature. For instance, an Italian study on sexual behaviour in the general population compared results obtained via computer assisted telephone interviews (CATI) with self-answered questionnaires following interviews (SAQ-FI) and found that the SAQ-FI sample reported higher levels of early intercourse and same-sex attraction and had lower item non-response rates than the CATI sample [17].

How these differing modes of survey administration affect data quality can be even more complicated with respect to research in migrant populations. In culturally and linguistically diverse settings, self-administered questionnaires (SAQ) (which tend to be written) may be problematic because "languages spoken may not have a standard written form, or respondent literacy rates may be vastly different" [18]. Likely reflective of such concerns, a recent review of 550 empirical surveys of asylum seekers and minority groups found that over half (n = 293) were administered through face-to-face interviews, compared to 11% (n = 55) SAQ [19].

When collecting sensitive data from potentially vulnerable populations, researchers have an ethical imperative to ensure that any foreseeable harms are proportionate to the benefits that can flow from valid and reliable research outputs. However, there is still no strong/empirical guidance to determine appropriate modes of SHBBV survey administration among migrant populations. Therefore, we aimed to perform a scoping review of SHBBV surveys administered to international migrant populations in receiver countries to understand the effect that mode of administration has on key indicators of data quality and reliability, including response rates and social desirability bias. Practical and logistical considerations associated with the different modes of administration were also considered. The PRISMA extension for scoping reviews has been followed in the reporting of this study [20].

## Materials and methods

An unregistered protocol was developed and is available on request from the corresponding author. The methodological framework for scoping reviews developed by Arksey and O'Malley [21] (set out in Table 1) was applied. The broad research objective was to determine what modes of survey administration have been used to conduct SHBBV surveys in migrant populations and to ascertain the strengths and limitations associated with each mode. The following sub-questions were set to meet the stated objective:

**Table 1. Methodological framework for scoping studies, based on Arksey and O'Malley [21].**

| STAGE | SUMMARY |
|---|---|
| 1. A research question is identified | Facets of the question (e.g. population, interventions, outcomes) are identified and defined. |
| 2. Potentially relevant studies are identified | A search strategy for a range of resources and databases is developed. The search is conducted within predefined parameters relating to factors such as language and date of publication/reporting. |
| 3. Relevant studies are selected | Studies identified in Stage Two are assessed against inclusion and exclusion criteria based on either a review of abstracts or the full study (if relevance cannot be established from the abstract). All studies which 'pass' this first screen are reviewed and assessed in full. |
| 4. Data are charted | Information relevant to the aims of the scoping review are extracted from each included study. |
| 5. Results are collated, summarized and reported | Data extracted in Stage Four are analyzed and findings are reported. |

1. With what frequency have different modes of administration been used to administer SHBBV surveys to migrants?

2. Is the mode of survey administration statistically associated with response rates, controlling for factors such as provision of recruitment incentives/gratuities and survey length?

3. What are the reported strengths and limitations of the different modes of survey administration, in terms of social desirability bias, project resources and other factors?

The review focussed on English-language papers published or released after 2000 (in light of the technological developments in survey administration). In order to be included, papers needed to contain: (a) primary analyses of data from SHBBV surveys administered to international migrants (i.e. people living in a country other than their birth country); and (b) some description of the mode of survey administration. General population surveys were excluded unless migrant and non-migrant responses were explicitly compared in the paper. Other exclusion criteria are set out in Table 2.

Searches were run in March and April 2019. The search strategy combined terms relating to three concepts–surveys, migrants and modes of survey administration. The terms were entered into the databases Embase, Medline and Web of Science (Core Collection) using database-specific subject headings and search syntax as set out in the Supplementary table (S1

**Table 2. Inclusion and exclusion criteria.**

| | Criteria |
|---|---|
| **Include if:** | • Study is primary analysis of survey data |
| | • Survey is aimed at or includes first-generation (externally-born) international migrants |
| | • Survey relates to the prevention, transmission or management of sexually transmissible infections and/or blood-borne viruses |
| | • Study contains some description of mode of survey administration |
| **Exclude if:** | • Survey is aimed at migrant health professionals only |
| | • Survey sample frame is the general population (unless the stated objective is to compare migrant and non-migrant responses) |
| | • Study relates to tourists / recreational travellers |
| | • Study seeks only to validate screening and diagnostic tools or tests used in clinical settings |
| | • Study is not in English language (although survey may be in another language) |

Table). Supplementary searches were conducted in Google, Google Scholar, and ProQuest Theses and Dissertations with a view to locating grey literature and unindexed publications. These supplementary searches were more focussed given the search limitations of those platforms (Survey AND (Migrant OR Refugee OR Displaced OR Emigrant OR Immigrant OR "Foreign born" OR "Culturally and Linguistically diverse") AND (STI OR STD OR BBV OR Sexual OR HIV OR "Hepatitis B" OR "Hepatitis C")). Only the first 20 pages of results in Google and Google Scholar were reviewed, consistent with accepted practice [22].

Results were imported into Endnote and de-duplicated using the process developed by Bramer, Giustini et al. [23] for this purpose (e.g. tailored use of field settings and filters). One researcher screened the title and abstract of each identified study against the inclusion and exclusion criteria set out in Table 2 and categorised each study as 'Potential Include' or 'Exclude'. The full text of all studies marked 'Potential Include' were then independently screened by two researchers and either marked for inclusion or exclusion with reasons. If the researchers reached different decisions, each researcher explained their rationale and, if consensus could not be reached, a third member of the research team assessed the item against the inclusion and exclusion criteria.

A charting form was developed in Excel by the second-named author and tested on the included studies identified through searches in Medline, Embase and Web of Science (see S2 Table). The form was revised for charting data in the remaining studies (i.e. those identified through Google, Google Scholar and ProQuest). The revisions involved reducing the number of charting categories and introducing fixed drop-down options into the Excel table (see S3 Table). Data were extracted by one researcher and cross-checked by a second researcher. Differences in coding decisions were resolved in the same manner as for screening (described above). The extracted data included information about the studies, including study setting, recruitment methods, sample size and characteristics, response rates, mode of survey administration, and reported information about the strengths and weaknesses of survey administration methods.

Following data extraction, it became apparent that different response rate calculation methods were being used in the included studies. As such, a decision was made to collect more detailed information relevant to response rate reporting. To that end, any studies in which a response rate was reported (or capable of being calculated) were reviewed and data extracted directly into Table 4 below, with a second researcher cross-checking for accuracy. Information on instrument validation was also collected *post facto* in response to a suggestion from one reviewer.

## Results

Ninety one studies were identified for inclusion following the search-and-screen process represented in Fig 1.

Key characteristics of the studies are set out in Table 3. The majority of studies (n = 51) were conducted in North America, followed by Europe/Eurasia (n = 22), Australia (n = 8), Asia (n = 6), Africa (n = 2) and Latin America and the Caribbean (LAC) (n = 2). Globally, the majority of SHBBV surveys were administered to migrants born in Asia (n = 40), LAC (n = 31) or Africa (n = 28). Only four included studies reported data from SHBBV surveys administered to migrants from Middle Eastern countries. Sample sizes ranged from six migrant participants [25] to 11,484 participants [26].

In 44 studies, existing SHBBV instruments were adapted or used. These instruments included the Brief HIV Screener [116], the Perceived Susceptibility to HIV Scale [117], the AIDS Health Belief Scale [118], the National Survey of Australian Secondary Students and Sexual Health [119], the Survey of Latino Adults [120, 121], the African Health and Sex Survey

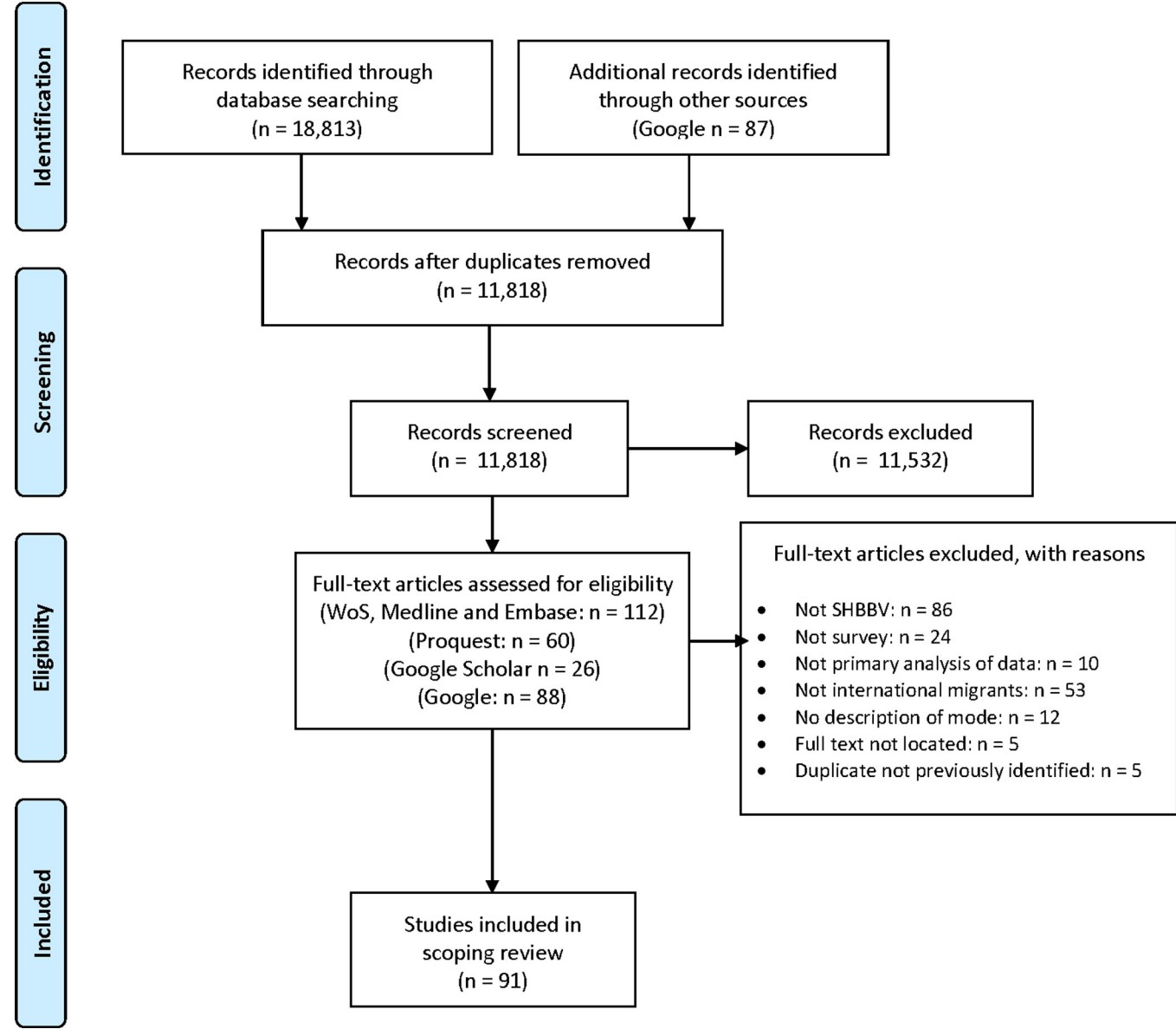

**Fig 1. PRISMA flow diagram of scoping review stages [24].**

[12], the Bass Line Survey [122], UN Behavioral Surveillance Surveys [123] and the National Health Interview Survey Supplement on AIDS Knowledge and Attitudes [124]. For the remaining studies, either the origin of the survey items was not reported (n = 23) or new survey instruments were developed (n = 24). Of the studies in which new survey instruments were developed, half (n = 12) did not explicitly report whether the instrument had been pretested or piloted.

As shown in Fig 2, 'interview only' was the most common mode of survey administration (n = 48), with face-to-face (n = 37) being the most common interview technique. Of the thirty six studies reporting data from 'self-completed' surveys only, pen-and-paper was the most common method of self-completion (n = 17). Few studies (n = 7) combined interview and self-completed methods of survey administration.

**Table 3. Included studies (n = 91), by key characteristics.**

| # | Author(s) and year | Country of study | Migrant region of origin | Mode of administration | SHBBV instrument used | Sample size |
|---|---|---|---|---|---|---|
| 1 | Agbemenu, Terry et al. [27] | USA | Africa | Paper | New instrument developed (not clear if tested) | 15 |
| | | | | Online | | |
| 2 | Ahmed [28] | USA | Africa | F2F interview | New instrument developed (not clear if tested) | 201 |
| 3 | Alber, Cohen et al. [29] | USA | Asia | Online | Based on / used an existing instrument | 418 |
| 4 | Alvarez-del Arco, Fakoya et al. [30] | Europe(9 countries) | Africa | Device | Based partly on / used an existing instrument | 2,209 |
| | | | LAC | | | |
| | | | Europe | | | |
| 5 | Amadi [31] | USA | Africa | Paper | Based partly on / used an existing instrument | 395 |
| 6 | Arevalo [32] | USA | LAC | Paper | Based on / used existing instrument | 80 |
| | | | | F2F interview | | |
| 7 | Asante, Körner et al. [33] | Australia | Africa Asia | Paper | New instrument developed (not clear if tested) | 286 |
| 8 | Bastani, Glenn et al. [34] | USA | Asia | Paper | Based on / used existing instruments | 1,123 |
| | | | | Phone interview | | |
| 9 | Beltran, Simms et al. [35] | USA | Asia | Online | Based on / used existing instruments | 192 |
| | | | | Paper | | |
| 10 | Burns, Fenton et al. [36] | UK | Africa | F2F interview | Based partly on / used existing instruments | 385 (incl. ~25% UK born) |
| | | | | Device | | |
| 11 | Chamratrithirong, Boonchalaksi et al. [37] | Thailand | Asia | F2F interview | New instrument developed + pretested | 3,426 |
| 12 | Chen, Guthrie et al. [38] | USA | Asia | Device | Based on / used existing instruments | 50 |
| 13 | Cohen [39] | USA | Asia | Paper | New instrument developed + pilot tested | 2,004 (excl. US born) |
| | | | | F2F interview | | |
| 14 | Coronado, Taylor et al. [40] | USA | Asia | F2F interview | New instrument developed + pretested | 430 (may include US-born) |
| 15 | Dean, Mitchell et al. [41] | Australia | Africa | Paper | Based on / used existing instruments | 229 |
| 16 | Delgado, Lundgren et al. [42] | USA | LAC | F2F interview | NR | 200 |
| 17 | Demeke [43] | USA | Africa | F2F interview | Based on / used existing instruments | 37 (excl. US born) |
| 18 | Dennis, Wheeler et al. [44] | USA | LAC | F2F interview | NR | 127 |
| 19 | Dias, Gama et al. [45] | Portugal | Africa | F2F interview | NR | 1,513 |
| | | | Asia | | | |
| | | | Europe | | | |
| | | | LAC | | | |
| 20 | Duan, Ding et al. [26] | China | Asia | F2F interview | Based on / used existing instruments | 11,484 |
| 21 | Elford, Doerner et al. [46] | UK | Africa | Online | NR | 1,334 |
| | | | Asia | | | |
| | | | Europe | | | |
| | | | LAC | | | |
| 22 | Elford, McKeown et al. [47] | UK | Africa | Online | Based on / used existing instruments | 1,241 |
| | | | Asia | | | |
| | | | Europe | | | |
| | | | LAC | | | |

*(Continued)*

**Table 3.** (*Continued*)

| # | Author(s) and year | Country of study | Migrant region of origin | Mode of administration | SHBBV instrument used | Sample size |
|---|---|---|---|---|---|---|
| 23 | Evans, Hart et al. [48] | UK | Europe | Online | NR | 206 |
| 24 | Evans, Suggs et al. [49] | UK | Africa | Paper | New instrument developed + pilot tested | 169 |
| | | | | Online | | |
| | | | | Phone | | |
| | | | | Device (SMS) | | |
| 25 | Fakoya, Alvarez-Del Arco et al. [50] | Europe (multiple) | Africa | Online | Based partly on / used existing instruments | 1,637 |
| | | | LAC | | | |
| 26 | Fenton, Chinouya et al. [51] | UK | Africa | Paper | New instrument developed (not clear if tested) | 720 (excl. UK born) |
| 27 | Fernandez-Esquer, Atkinson et al. [52] | USA | LAC | F2F interview | Based partly on / used an existing instrument | 152 |
| 28 | Fitzgerald, Chakraborty et al. [53] | USA | LAC | F2F interview | New instrument developed (not clear if tested) | 19 (excl. US born) |
| 29 | Ford and Chamrathrithirong [54] | Thailand | Asia | F2F interview | New instrument developed + pretested | 3,426 |
| 30 | Foster, McCormack et al. [55] | Australia | Asia | Paper | Based on / used instruments | 435 |
| 31 | Getrich, Broidy et al. [56] | USA | LAC | F2F interview | NR | 6 (excl. US-born) |
| 32 | Goldade and Nichter [57] | Costa Rica | LAC | F2F interview | NR | 33 |
| 33 | Gray, Crawford et al. [58] | Australia | Africa | Paper | Based on / used existing instruments | 209 |
| | | | Asia | Device | | |
| | | | | Online | | |
| 34 | Grieb, Flores-Miller et al. [59] | USA | LAC | Paper | NR | 104 |
| 35 | Hamdiui, Stein et al. [60] | Netherlands | Africa | Paper | New instrument developed + pretested | 193 (excl. Dutch born) |
| | | | | Online | | |
| 36 | Hislop, Teh et al. [61] | Canada | Asia | F2F interview | New instrument developed + pretested | 503 |
| 37 | Hwang, Huang et al. [62] | USA | Asia | Paper | Based on / used existing instruments | 128 (excl. US born) |
| 38 | Jenkins, McPhee et al. [63] | USA | Asia | Phone interview | New instrument developed + pretested | 1508 |
| 39 | Johnston [64] | Armenia | Europe | F2F interview | New instrument developed + piloted | 945 |
| | | Azerbaijan | | | | |
| | | Georgia | | | | |
| 40 | Joseph, Belizaire et al. [65] | USA | LAC | F2F interview | New instrument developed (not clear if tested) | 20 (excl. US born) |
| 41 | Juon, Strong et al. [66] | USA | Asia | Paper | NR | 232 |
| 42 | Juon, Lee et al. [67] | USA | Asia | Paper | NR | 877 |
| 43 | Kara [68] | USA | Africa | Paper | Based on / used existing instruments | 164 |
| | | | | Online | | |
| 44 | Kuehne, Koschollek et al. [69] | Germany | Africa | Paper | Based on / used existing instruments | 2,720 |
| | | | | F2F interview | | |
| 45 | Leite, Buresh et al. [70] | USA | LAC | F2F interview | New instrument developed (not clear if tested) | 200 (excl. US born) |
| 46 | Lessard, Lebouche et al. [71] | Canada | Africa | Phone interview | Based on / used existing instruments | 40 |
| | | | Asia | | | |
| | | | Europe | | | |
| | | | LAC | | | |
| | | | Middle East | | | |

(*Continued*)

**Table 3.** (Continued)

| # | Author(s) and year | Country of study | Migrant region of origin | Mode of administration | SHBBV instrument used | Sample size |
|---|---|---|---|---|---|---|
| 47 | Lin, Simoni et al. [72] | USA | Asia | Online | Based partly on / used existing instruments | 144 |
| 48 | Lofters, Vahabi et al. [73] | Canada | Asia | Paper | NR | 30 |
| 49 | Loos, Manirankunda et al [74] | Belgium | Africa | Paper | NR | 139 |
| | | | LAC | | | |
| 50 | McGregor, Mlambo et al. [13] | Australia | Africa | Paper | Based on / used existing instruments + pilot tested | 1,406 |
| | | | Asia | | | |
| 51 | Manoyos, Tangmunkongvorakul et al. [75] | Thailand | Asia | F2F interview | Based on / used existing instruments | 442 |
| 52 | Maxwell, Bastani et al. [76] | USA | LAC | F2F interview | Based partly on / used existing instruments | 106 |
| | | | | Phone interview | | |
| 53 | Miller, Guarnaccia et al. [77] | USA | LAC | Phone interview | Based on / used existing instruments | 85 (excl. US born) |
| 54 | Montealegre [78] | USA | LAC | F2F interview | NR | 210 |
| 55 | Montealegre, Risser et al. [79] | USA | LAC | F2F interview | NR | 210 |
| 56 | O'Connor, Shaw et al. [80] | Australia | Asia | Phone interview | Based on / used existing instruments | 499 |
| 57 | Ogungbade [81] | USA | Africa | Paper | Based on / used existing instruments | 167 |
| 58 | Organista and Kubo [82] | USA | LAC | F2F interview | Based on / used existing instruments | 102 |
| 59 | Pannetier, Ravalihasy et al. [83] | France | Africa | F2F interview | Based on / used existing instruments | 980 |
| 60 | Platt, Grenfell et al. [84] | UK | Europe | Device | NR | 268 |
| 61 | Plewes, Lee et al. [85] | Thailand | Asia | F2F interview | NR | 109 |
| 62 | Ramanathan and Sitharthan [86] | Australia | Asia | Online | Based on / used existing instruments | 184 |
| 63 | Rangel, Martinez-Donate et al. [87] | Mexico | LAC | Paper | New instrument developed (not clear if tested) | 1,429 |
| 64 | Saenz [88] | USA | LAC | F2F interview | Based on / used existing instruments | 141 |
| 65 | Salabarria-Pena, Lee et al. [89] | USA | LAC | F2F interview | New instrument developed (not clear if tested) | 175 |
| 66 | Salehi [90] | Canada | Various (unspecified) | Paper | Based on / used existing instruments | 141 |
| 67 | Santos-Hovener, Marcus et al. [91] | Germany | Africa | Paper | Based on / used existing instruments + pretested | 596 |
| | | | | F2F interview | | |
| | | | | Phone interview | | |
| 68 | Selvey, Lobo et al. [92] | Australia | Asia | Paper | Based on / used existing instruments | 94 (excl. non-Asian born) |
| | | | | Online | | |
| 69 | Shiau, Bove et al. [93] | USA | Asia | F2F interview | New instrument developed (not clear if pretested) | 270 (excl. US born) |
| | | | | Phone interview | | |
| 70 | Şimşek, Yentur Doni et al. [94] | Turkey | Middle East | F2F interview | Based on / used existing instruments | 458 |
| 71 | Spadafino, Martinez et al. [95] | USA | LAC | F2F interview | NR | 176 |
| | | | | Phone interview | | |
| 72 | Srithanaviboonchai, Choi et al. [96] | Thailand | Asia | F2F interview | NR | 429 |

(*Continued*)

**Table 3.** (Continued)

| # | Author(s) and year | Country of study | Migrant region of origin | Mode of administration | SHBBV instrument used | Sample size |
|---|---|---|---|---|---|---|
| 73 | Stromdahl, Liljeros et al. [97] | Sweden | Africa | Online | New instrument developed + piloted | 244 |
| | | | Asia | | | |
| | | | Europe | | | |
| | | | LAC | | | |
| 74 | Sumari-de Boer, Sprangers et al. [98] | Netherlands | Africa | F2F interview | Based on / used existing instruments | 112 |
| | | | Europe | | | |
| 75 | Taylor, Jackson et al. [99] | USA | Asia | Phone interview | Based on / used existing instruments | 75 |
| 76 | Taylor, Jackson et al. [100] | USA | Asia | F2F interview | Based on / used existing instruments | 413 |
| 77 | Taylor, Choe et al. [101] | USA | Asia | F2F interview | Based on / used existing instruments | 715 |
| 78 | Taylor, Tu et al. [102] | USA | Asia | F2F interview | New instrument developed + pretested | 395 |
| 79 | Taylor, Seng et al. [103] | USA | Asia | Phone interview | NR | 111 |
| 80 | Thompson, Taylor et al. [104] | USA | Asia | F2F interview | Based on / used existing instruments | 116 (excl. North American born) |
| 81 | Tu, Li et al. [105] | USA | Asia | F2F interview | New instrument developed (not clear if pretested) | 945 (excl. USA and Can. born) |
| | | Canada | | | | |
| 82 | UNHCR [106] | Zambia | Africa | F2F interview | Based on / used existing instruments | 822 |
| 83 | UNHCR [107] | Kenya | Africa | F2F interview | Based on / used existing instruments | 1,646 |
| 84 | Uribe, Darrow et al. [108] | USA | LAC | Phone | NR | 1,266 (excl. US born) |
| 85 | van der Veen, Voeten et al. [109] | Netherlands | Middle East | Paper | Based partly on / used existing instruments | 174 (excl. Dutch born) |
| 86 | Viadro and Earp [110] | USA | LAC | F2F interview | NR | 43 |
| 87 | Villarreal, Wiley et al. [111] | USA | LAC | Paper | New instrument developed + piloted | 24 (excl. US born) |
| 88 | Westmaas, Kok et al. [112] | Netherlands | Europe | Paper | Based on / used existing instruments | 753 |
| | | | | Online | | |
| 89 | Yau, Ford et al. [113] | Canada | Asia | Phone interview | New instrument developed (not clear if tested) | 1,013 overall (may include Canadian born) |
| 90 | Zellner, Martínez-Donate et al. [114] | USA | LAC | Device | NR | 647, excl. US born |
| 91 | Zhussupov, McNutt et al. [115] | Kazakhstan | Middle East | F2F interview | NR | 422 |

F2F = face-to-face

NR = not reported

LAC = Latin America and the Caribbean

NR = Not reported

Fig 3 shows that some modes of SHBBV survey administration have been implemented more in some populations, compared to others. For instance, face-to-face only interviews were more commonly administered to LAC (n = 16) and Asian (n = 13) migrants; by contrast, pen-and-paper only surveys were used less frequently in LAC communities (n = 4). The small number of 'online only' and 'device only' surveys were relatively evenly distributed across LAC, Asian, African and European migrant populations.

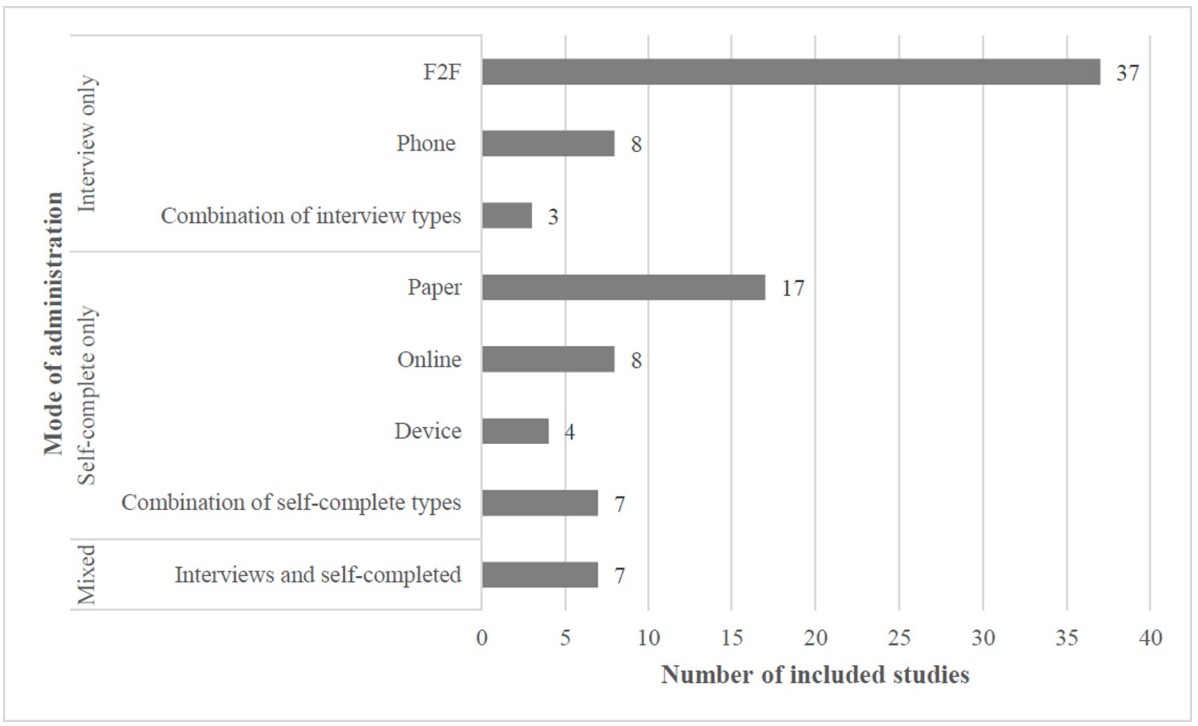

**Fig 2. Included studies (n = 91), by mode of survey administration.**

Given the level of reporting in the included studies, it was not possible to determine whether certain modes of administration were associated with higher response rates,

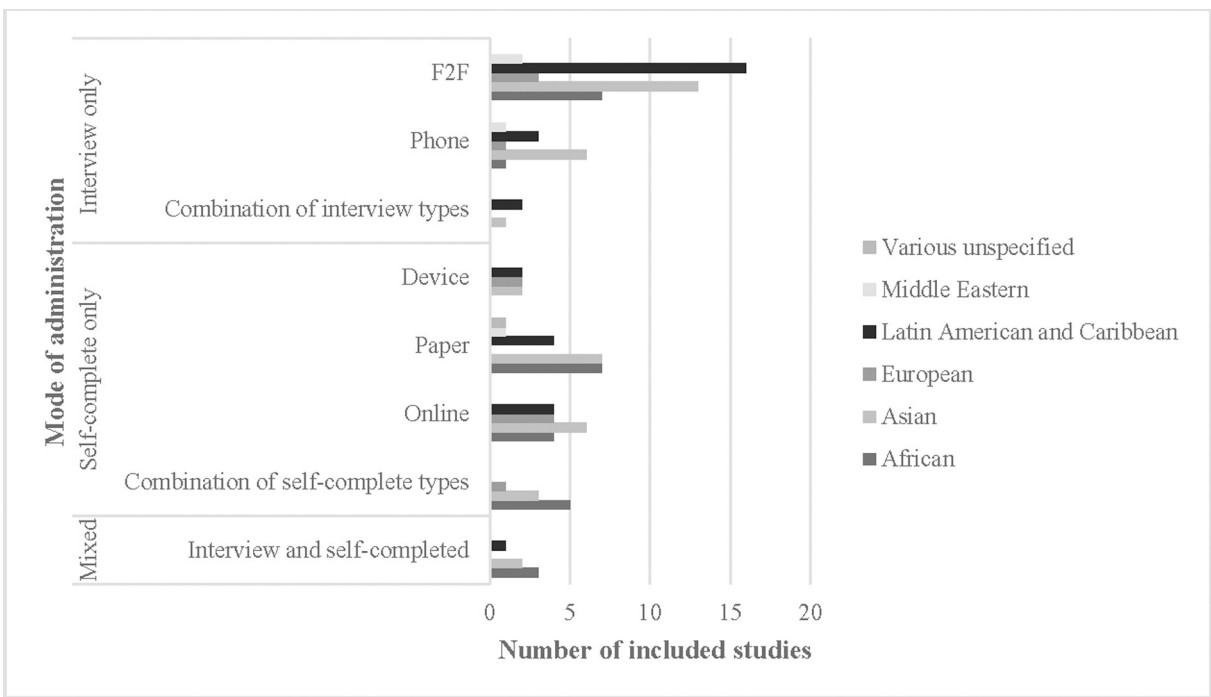

**Fig 3. Included studies (n = 91), by mode of administration and migrants' region of origin.**

controlling for other factors. Sixty one studies (67%) either: (a) did not report response rates or the data necessary to calculate response rates; or (b) partially reported them (e.g. did not specifically report for overseas-born sample members or did not report response rates for all modes of administration).

Of the 30 studies where response rates were reported or able to be calculated (Table 4), the most common mode of administration was face-to-face interview only (n = 12), followed by pen-and-paper only (n = 6). By contrast, online administration was only used in four studies and was used in combination with other modes in three of those cases. Several studies noted the difficulties associated with determining the denominator required to calculate response rates when administering surveys online (e.g. Elford, Doerner et al. [46], Fakoya, Alvarez-Del Arco et al. [50], Gray, Crawford et al. [58]). Additionally, Ramanathan and Sitharthan [86] noted that noneligible persons could participate in online surveys and that the same respondent could complete the survey multiple times unless identifying data (e.g. IP addresses) were collected and stored.

The majority of the 30 studies reported response rates exceeding 50 percent, although the methods for calculating response rates varied. Generally, response rates were calculated by dividing the number of complete (and, in some cases, partial) eligible surveys by the sum of the number of ineligible cases, refusals, unsuccessful contact attempts and all cases of unknown eligibility. However, in other studies, attempts were made to estimate the number of cases of unknown eligibility which were ineligible and those cases were excluded from the denominator. For instance, Taylor, Choe et al. [101] reported "the overall estimated response rate was 80% among men and 82% among women (assuming the same proportions of eligible men and women among those who could and could not be contacted)".

Additionally, there was a general lack of reporting on data relevant to assessing the quality of the response rates. For instance, 14 studies (47%) did not report whether incentives to participation were offered, 14 studies (47%) did not report on the method for obtaining informed consent, and 27 studies (90%) did not provide data to enable the characteristics of participants to be compared to non-responders.

The strengths and limitations of the various methods were discussed in the examined literature. Face-to-face interviews were considered useful when surveying populations with low levels of literacy [52, 89]. However, this method of administration was often human resource intensive and associated with logistical issues, including the need to find accessible and sufficiently private interview sites at mutually convenient times for multiple researchers and participants [64, 78, 79, 106].

While one study considered face-to-face interaction to be an important element of building trust [70], several noted the potentially increased risk of social desirability bias when disclosing sensitive information in-person [32, 35, 44, 51, 52, 83, 89, 110], and it was noted that consideration should be given to the characteristics of the interviewer. For instance, in a study of the health knowledge, attitudes and behaviours of U.S. Latino men who have sex with men, Arevalo [32] warned that "[s]ocial desirability may have been magnified by the interviewer, given that he was relatively more verbal, educated and overall healthier than the average participant." A study of the sexual behaviour of male Mexican migrants to the United States observed that extramarital sex may have been underreported to female interviewers who lived in the respondents' community or were casually known to the respondents' wives [110].

Although telephone interviews have the potential to increase respondents' sense of 'anonymity', the risk of social desirability bias may still remain because telephone respondents might be unable to answer questions in a private location away from other household residents [108]. The included studies also report that telephone interviews may result in selection bias [40, 63, 77, 100, 113]. For instance, Miller, Guarnaccia et al. [77] observed that recent Latino

**Table 4. Included studies with response rates reported or able to be calculated (n = 30), by other reported characteristics.**

| Study | Survey mode | Reported response rate (%) | Reported information relevant to RR | How many and what type of attempts were made to contact subjects?* | Who approached potential subjects?* | Where were potential subjects approached?* | How was informed consent obtained?* | How did those who agreed differ from those who did not agree?* | What was the average time taken to complete survey (minutes)? | Was an incentive to participation offered? |
|---|---|---|---|---|---|---|---|---|---|---|
| Ahmed [28] | F2F interview | 48 | Number invited: 425 Number consented: 205 | NR | NR | Mosques, local cafés, restaurants | Verbal | NR | 60 | No |
| Alvarez-del Arco, Fakoya et al. [30] | Device | 70 | Number invited: 3251 Number eligible: 3152 Number consented: 2209 | NR | Researcher or member of clinical care team | Clinic | NR | Participation higher in men, decreased with age and was higher in migrants from Latin America and Eastern Europe and lower in those from Sub-Saharan Africa | NR | NR |
| Asante, Körner et al. [33] | Paper | >95 in Thai, Ethiopian and Sudanese communities Less in Cambodian community | Only reported for Cambodian community Number invited: 104 Number consented: 86 | NR | Co-workers from the relevant language backgrounds, as well as some members of the reference groups, would lead the recruitment and assist participants to complete the questionnaires | Mainly at places of worship, community events and other social gatherings | NR | NR | 20–25 | NR |
| Bastani, Glenn et al. [34] | Paper Phone interview | 94 (B) 86 (F) | Number screened: 1,866 Number eligible: 1,196 Number enrolled at baseline: 1,123 | In-person on a single day | Staff members | Church | NR | NR | NR | NR |
| Coronado, Taylor et al. [40] | F2F interview | Figure not reported but relevant data presented (see next column) | Number screened: 1,902 Number enrolled: 436 (6 later excluded due to ineligibility) Number refused: 314 Number ineligible at screening: 628 households + 105 (non-residential) Number unable to be contacted: 419 | Households received an introductory letter (traditional Chinese, simplified Chinese, and English versions). Called 2 weeks later. Up to 5 contact attempts made. | Chinese interviewer of same gender | Home | NR | NR | 30 | Calendar and $20 |

*(Continued)*

Table 4. (Continued)

| Study | Survey mode | Reported response rate (%) | Reported information relevant to RR | How many and what type of attempts were made to contact subjects?* | Who approached potential subjects?* | Where were potential subjects approached?* | How was informed consent obtained?* | How did those who agreed differ from those who did not agree?* | What was the average time taken to complete survey (minutes)? | Was an incentive to participation offered? |
|---|---|---|---|---|---|---|---|---|---|---|
| Duan, Ding et al. [26] | F2F interview | Figure not reported but relevant data presented (see next column) | Among the total of 7656 mixed couples, 6269 Chinese spouses and 7092 Burmese immigrant spouses gave informed consent to participate including both spouses of 5742 couples. Only the 5742 couples with both spouses participating in the survey were included. | NR | Trained public health worker (or, where necessary, village or community clinical doctors who were able to speak and understand Burmese) | NR (but interviews principally conducted in homes) | NR | NR | NR | $10 |
| Evans, Suggs et al. [49] | Paper Online Phone Device (SMS) | 61 | Number invited: 281 Number consented: 172 (3 later excluded) | Not reported for baseline | Community researchers | Voluntary sector groups and community venues | Written | NR | NR | GBP 5 shopping voucher |
| Foster, McCormack et al. [55] | Paper | 94 | Number distributed: 488 Number returned: 460 | NR | Sexual health clinic staff and health education officers | Sexual health clinic and sex work parlours | Implied (consent form prefaced survey) | NR | NR | NR |
| Grieb, Flores-Miller et al. [59] | Paper | Figure not reported but relevant data presented (see next column) | Number screened: 135 Number eligible: 113 Number consented: 104 | NR | Trained research assistants | Street- and community-based venues | Verbal | NR for non-response/refusals but noted that no differences in age, country of origin, education, time in the United States, or time in Baltimore were observed between those who were eligible and those who were not. | 10–15 | $10 |

(Continued)

Table 4. (Continued)

| Study | Survey mode | Reported response rate (%) | Reported information relevant to RR | How many and what type of attempts were made to contact subjects?* | Who approached potential subjects?* | Where were potential subjects approached?* | How was informed consent obtained?* | How did those who agreed differ from those who did not agree?* | What was the average time taken to complete survey (minutes)? | Was an incentive to participation offered? |
|---|---|---|---|---|---|---|---|---|---|---|
| Hamdiui, Stein et al. [60] | Paper Online | 69.1 | Number invited: 350 Number participated: 242 (excluding 165 people recruited by participants who accepted the invitation–no RR reported for the total sample which included respondent-driven sampling methods) | Number of attempts NR Online-recruited respondents were enrolled through advertisements on Moroccan-Dutch forums, Facebook, Instagram, websites. Recruiting peers online was enabled through indirect email, WhatsApp, Facebook, or by sharing a hyperlink. | Peer (respondent-driven sampling) | Online and at community venues, such as community centres, day care centres, mosques, interest groups, and civil support foundations. | NR | NR | NR | Gift coupon when recruited at least three other respondents (value increased in three steps to enhance peer recruitment: €5, €10, and €25). |
| Hislop, Teh et al. [61] | F2F interview | 59 | Households selected: 1500 Number of non-residential: 41 Number unable to contact: 149 Number ineligible: 375 Number refused: 384 Number of interviews completed: 551 (504 when non-migrants excluded) | Five door-to-door attempts | Trilingual Chinese interviewer | Home | NR | NR | 45 | $20 |
| Jenkins, McPhee et al. [63] | Phone interview | 93 | Call attempts: 12,094 Call attempts that reached eligible respondents: 1624 Number consented: 1508 | Up to five attempts by phone | NR | Phone | NR | NR | NR | NR |

(Continued)

Table 4. (Continued)

| Study | Survey mode | Reported response rate (%) | Reported information relevant to RR | How many and what type of attempts were made to contact subjects?* | Who approached potential subjects?* | Where were potential subjects approached?* | How was informed consent obtained?* | How did those who agreed differ from those who did not agree?* | What was the average time taken to complete survey (minutes)? | Was an incentive to participation offered? |
|---|---|---|---|---|---|---|---|---|---|---|
| Juon, Lee et al. [67] | Paper | 98 (B) 78 (F) | Eligible program participants: 940 Number of no-shows: 47 Number who did not complete baseline or did not participate: 13 Number who participated in program in past year: 3 Number who completed baseline: 877 Number who completed follow-up: 688 | NR | NR | Community based organisations, college cultural organisations, Asian grocery stores, restaurants, nail salons | NR | NR for baseline Differences at follow-up described | NR | NR |
| Kara [68] | Paper Online | 35 | Number of surveys distributed: 525 Number of surveys returned: 186 | NR | Partners from member organisations made initial contact | Online | Electronic (for online survey) Implied (for written survey) | NR | 10–30 | NR |
| Lessard, Lebouche et al. [71] | Phone interview | 54 | Number eligible: 74 Number refused: 4 Number unable to be contacted: 30 Number participants: 40 | NR | Service staff member made initial contact, followed up by researcher | Phone | Written | NR | 10–15 | None |
| Maxwell, Bastani et al. [76] | F2F interview Phone interview | 51 (B) 68 (F) | Number recruited at clinic: 98 Number who attended workshop: 46 (+ 8 peer recruits) Number who completed baseline survey: 54 Number who completed follow-up survey: 44 workshop attenders and 28 non-attenders | NR | Clinical phlebotomist briefly described study, researcher followed up with those interested | Clinic | Written | Participants who completed post-test reported significantly more years of schooling than those who did not complete. | NR | $5 for initial interview, $10 for workshop participation, $15 for post-test |

(Continued)

Table 4. (Continued)

| Study | Survey mode | Reported response rate (%) | Reported information relevant to RR | How many and what type of attempts were made to contact subjects?* | Who approached potential subjects?* | Where were potential subjects approached?* | How was informed consent obtained?* | How did those who agreed differ from those who did not agree?* | What was the average time taken to complete survey (minutes)? | Was an incentive to participation offered? |
|---|---|---|---|---|---|---|---|---|---|---|
| Montealegre, Risser et al. [79] | F2F interview | Figure not reported but relevant data presented (see next column) | Number screened: 230 Number eligible: 222 Number consented: 221 (one did not complete interview and data from ten excluded from analysis or lost) | Number of attempts NR Seeds and eligible participants were given three serially numbered study coupons to recruit peers. Study coupons provided recruits with the name and a short description of the study, project phone number, name and address of the interview sites, hours of operation, and the coupon's expiration date. | Peer (respondent-driven sampling) | NR | Verbal | NR | 60 | Seeds and participants were given $20 for completing the interview and $5 for each of up to three peers they recruited into the survey. |
| O'Connor, Shaw et al. [80] | Phone interview | 66 | Number invited: 761 Number consented: 506 (seven later excluded from analysis) | NR | Men were telephoned by a Vietnamese speaking woman | Phone | NR | NR | NR | NR |
| Ogungbade [81] | Paper | 86 | Number of surveys distributed: 194 Number of surveys returned: 167 | Flyers distributed. Researcher addressed potential participants at an event. Returned one week later to conduct survey. | Researcher (Nigerian migrant) | Faith-based organisations | Implied consent form given explaining that completion of survey was considered consent | NR | NR | NR |
| Organista and Kubo [82] | F2F interview | >90 | Notes from outreach workers indicate that less than 10% of men approached refused participation. | NR | Spanish-speaking project team members who introduced themselves as local public health outreach workers | Street corner | NR | NR | 45 | $20 fast food voucher |

(Continued)

Table 4. (Continued)

| Study | Survey mode | Reported response rate (%) | Reported information relevant to RR | How many and what type of attempts were made to contact subjects?* | Who approached potential subjects?* | Where were potential subjects approached?* | How was informed consent obtained?* | How did those who agreed differ from those who did not agree?* | What was the average time taken to complete survey (minutes)? | Was an incentive to participation offered? |
|---|---|---|---|---|---|---|---|---|---|---|
| Ramanathan and Sitharthan [86] | Online | 42 | Number surveys attempted: 438 Number of surveys completed: 278 Number of completed surveys in which SHBBV section also completed: 184 | Advertisements on Indian specific websites and social networking websites (e.g. Google, Facebook). | N/A (internet advertising) | Indian specific websites and social networking websites | NR | NR | NR | NR |
| Rangel, Martinez-Donate et al. [87] | Paper | 90 | Number invited: 1,606 Number consented: 1,429 | NR | Trained Mexican interviewers | International airport, bus stations, deportation stations | Verbal | NR | NR | NR |
| Salabarria-Pena, Lee et al. [89] | F2F interview | 97 | Number invited: 222 Number ineligible: 42 Number refused: 5 | In-person (quantity unclear) | NR | Clinic waiting room | Verbal | NR | 60 | NR |
| Santos-Hovener, Marcus et al. [91] | Paper F2F interview Phone interview | Figure not reported but relevant data presented (see next column) | Surveys distributed: 950 Number returned: 649 Number eligible: 569 | NR | Peer researchers | NR | Verbal | NR | NR | Key chain, shopping cart chip, referral to health promotion information sessions, condom, informational flyers and free testing services |
| Şimşek, Yentur Doni et al. [94] | F2F interview | 100 | A total of 961 married women were identified in 458 houses. One eligible woman from each selected house was randomly selected. A total of 458 women provided written and signed informed consent; the response rate among eligible women was 100.0 percent. | 12 attempts made to contact | Trained Syrian midwife research assistant, lab technician and a translator from the area | Home | Written and verbal | NA | NR | NR |

(Continued)

**Table 4.** (Continued)

| Study | Survey mode | Reported response rate (%) | Reported information relevant to RR | How many and what type of attempts were made to contact subjects?* | Who approached potential subjects?* | Where were potential subjects approached?* | How was informed consent obtained?* | How did those who agreed differ from those who did not agree?* | What was the average time taken to complete survey (minutes)? | Was an incentive to participation offered? |
|---|---|---|---|---|---|---|---|---|---|---|
| Taylor, Jackson et al. [99] | Phone interview | 70 | Initial sampling frame: 161 Number unable to contact/ phone disconnected: 42 Number ineligible: 12 Number consented: 75 | Number of attempts NR Introductory letter followed by telephone call | Bilingual, bicultural Vietnamese survey workers | Home | NR | NR | NR | $10 voucher |
| Taylor, Jackson et al. [100] | F2F interview | 73 (B) 56 (F) | NR for baseline Three hundred and twenty (77 percent) of the 413 women who participated in the baseline survey also completed the follow-up survey. Therefore, the estimated overall response rate with respect to the hepatitis B questions was 56 percent (i.e., 77 percent of 73 percent). | NR | Bilingual, bicultural Cambodian women | Home | NR | NR | NR | Calendar at baseline, $5 at follow up |
| Taylor, Choe et al. [101] | F2F interview | 80–82 | Details obtained from related papers cited. Number of unsuccessful contact attempts: 41 (women); 47 (men) Number ineligible: 116 (women); 131 (men) Number eligible but refused: 66 (women); 70 (men) Number completed: 370 (women); 345 (men) | Five door-to-door attempts | Bilingual, bicultural interviewers (gender matched) | Home | NR | NR | 45 | Posters |

(Continued)

Table 4. (Continued)

| Study | Survey mode | Reported response rate (%) | Reported information relevant to RR | How many and what type of attempts were made to contact subjects?* | Who approached potential subjects?* | Where were potential subjects approached?* | How was informed consent obtained?* | How did those who agreed differ from those who did not agree?* | What was the average time taken to complete survey (minutes)? | Was an incentive to participation offered? |
|---|---|---|---|---|---|---|---|---|---|---|
| | | | Estimated proportion of eligible where eligibility was not established: 79% (women); assume proportion of eligible same as those not contactable (men) | | | | | | | |
| Taylor, Tu et al. [102] | F2F interview | Figure not reported but relevant data presented (see next column) | Interviews completed: 436 Number of households refused: 314 Number of households ineligible: 628 Number of uncontactable households: 419 (plus 105 non-residential addresses) | Introductory letter followed by five door-to-door attempts | Chinese interviewer of same gender | Home | NR | NR | NR | $20 |
| UNHCR [106] | F2F interview | Figure not reported but relevant data presented (see next column) | Kala camp Number of forms completed: 828 Number of refusals: 34 Number unable to contact: 224 Kala communities Number of forms completed: 880 Number of refusals: 17 Number unable to contact: 169 | In-person 1–3 times | Research assistants | Home | Oral (with interviewer's signature) | NR | NR | NR |

(*Continued*)

**Table 4.** (Continued)

| Study | Survey mode | Reported response rate (%) | Reported information relevant to RR | How many and what type of attempts were made to contact subjects?* | Who approached potential subjects?* | Where were potential subjects approached?* | How was informed consent obtained?* | How did those who agreed differ from those who did not agree?* | What was the average time taken to complete survey (minutes)? | Was an incentive to participation offered? |
|---|---|---|---|---|---|---|---|---|---|---|
| | | | Mwange camp Number of forms completed: 916 Number of refusals: 20 Number unable to contact: 389 Mwange communities Number of forms completed: 854 Number of refusals: 16 Number unable to contact: 349 | | | | | | | |

migrants to New Jersey were less likely to have residential telephones or may have "rapid turn-over of telephone numbers" due to high residential mobility.

The risk of selection bias was also reported in the studies which utilised online surveys [29, 49, 92, 97]. Online surveys have the potential to exclude respondents who lack internet access or technological proficiency, or who are wary of disclosing sensitive information online. Selvey, Lobo, et al. [92] found that only a minority of Asian sex workers in Australia completed online versions of a survey, with most preferring pen-and-paper (although the difference may have been attributable to the recruitment methods associated with each). A study of HIV testing among African migrants living in the UK found that none of the 169 respondents completed an online follow-up survey, although 60 subsequently agreed to participate in a telephone interview; this led the authors to conclude that online data collection "was not feasible in this population group" [49]. However, online recruitment and administration was considered advantageous in studies of migrant men who have sex with men (MSM) in Britain [46, 47]. According to Elford, McKeown, et al. [47], "[u]sing an online survey we were able to survey MSM across Britain from a diverse range of backgrounds."

One study recommended the use of computer-assisted self-interviews (CASI) in future research as a means of "address[ing] the need for privacy and the low literacy levels" in some migrant populations [52]. Empirical data on the strengths and limitations of this mode of survey administration were not presented in any of the included studies.

## Discussion

The primary objective of this scoping review was to determine best practices from the published literature to ensure that future SHBBV surveys are conducted both effectively and efficiently in migrant populations. However, the widespread lack of reporting on key survey characteristics made it difficult to appraise which mode of survey administration is likely to collect the most reliable data to inform future migrant SHBBV service provision and planning. Researchers are thus limited in their ability to avoid past missteps and replicate successes in study design, creating the risk of both resources and participants' time being wasted.

Only a minority of studies in this scoping review reported response rates and, of those, few provided a comprehensive description of other key survey characteristics. The findings are consistent with a recent review of empirical surveys of asylum-related migrants and minority groups which found that "information on methodological aspects, such as response/cooperation/participation rate, sampling frames, sampling strategies . . . are often missing or are not specified and discussed" [19]. These findings emphasise the need for greater adherence to (or awareness of) reporting standards [125]. For instance, the STROBE checklist for observational studies requires details about setting (e.g. recruitment sites and sources), eligibility criteria, method of recruitment, and numbers of individuals at each stage of the study, and reasons for non-participation at each stage [126]. Survey-specific checklists also recommend reporting additional details including description of the survey instrument and its development, pretesting processes, instrument reliability and validity, sample representativeness, mode of administration, number of attempts made to contact subjects, whether incentives were offered, methods for analysis of nonresponse error and descriptions of consent procedures (see data extraction tool published by Bennett, Khangura, et al. [127]).

There are also ethical implications associated with the lack of transparency. The principles of beneficence and non-maleficence require researchers "to seek the greatest benefit for research participants while minimizing harm" [128]. When examining a sensitive subject matter (e.g. sexual knowledge and behaviours) with potentially vulnerable participants (e.g. migrants), researchers must feel confident that any *potential discomfort* to participants is

outweighed by the *expected benefits* which, at the very least, should take the form of valid and reliable findings. In order to weigh the potential harms against the potential benefits, researchers need to understand how SHBBV information has been collected from migrant populations in the past, and whether those methods produced valid and reliable data (and, if not, why not). This ethical arithmetic is not easily performed based on the information reported in the studies included in this review.

The information that we have about the use of online SHBBV surveys in migrant populations offers a case in point. There has been an increase in the use of online surveys for SHBBV research in migrant populations since 2010, as is evident in Table 3; this reflects increased general access to the internet and the development of a number of affordable and accessible survey software development tools [129, 130]. However, the included studies in this scoping review broach some important considerations about the appropriateness of online SHBBV surveys in migrant settings. For instance, Selvey, Lobo et al. [92] and Evans, Suggs et al. [49] had limited success in using online surveys to obtain data from Asian Australian sex workers and African migrants living in the UK respectively. By contrast, Elford et al. [46, 47] considered SHBBV online surveys a useful tool. Based on the available information, future researchers are faced with a dilemma as to whether they can reasonably expect valid and reliable SHBBV data from online surveys in migrant populations. More data are needed to provide guidance to researchers considering the use of this mode of survey administration. As Poynton, DeFouw, et al. [131] note, online survey methods "will continue to be poorly understood until researchers plan for and more thoroughly report information related to response rates." Their specific recommendations for the conduct and reporting of online survey research should be heeded (e.g. create separate links to the survey for each mode of invitation or dissemination; document undeliverable emails; keep records of the number of people on electronic mailing lists and in online discussion boards) [131].

Despite the dearth of data reported in the included studies, the following principles are suggested to guide the administration of SHBBV surveys in migrant contexts:

1. SHBBV survey researchers should begin the survey design process with a clear profile of their sample population. The profile can either be created by drawing upon existing data or in consultation with informed community stakeholders. Where possible, the profile should include information about: (a) languages spoken; (b) written literacy; (c) access to relevant technology (e.g. internet, phones) and technological proficiency; (d) social customs governing researcher/participant interactions (e.g. gender/class considerations); and (e) perceived attitudes to the subject areas that are the focus of the survey.

2. The sample profile should inform the choice of survey administration mode, based on mode-specific considerations which include those set out in Table 5.

3. Where possible, consideration should be given to mixed-modes of survey data collection to overcome the limitations associated with using each mode in isolation [19]. However, mixed-mode survey administration is not, in itself, a magic bullet and care still needs to be taken to avoid measurement (and other) errors that may affect the validity and reliability of the findings [132, 133].

4. In the absence of clear guidance on best practice in the administration of SHBV surveys in specific migrant populations, pre-testing and pilot testing are essential. Pre-testing will enable "the capabilities of the selected mode[s] of data collection" to be evaluated, while pilot testing can be used to estimate response rates and ascertain whether a proposed mode of administration is appropriate for meeting research objectives [134]. Where issues are identified through pre-testing and piloting, appropriate revisions should be made in line

**Table 5. Advantages, disadvantages and considerations, by mode survey administration.**

| Mode | Possible advantages | Possible disadvantages | Considerations and significance | |
|---|---|---|---|---|
| Self-completed | • Versions of the survey can be prepared in multiple languages<br>• Allows for greater anonymity which can reduce social desirability response bias, especially when asking sensitive questions<br>• Can be completed at participants' own convenience and does not have to be completed in full in one sitting | • Less control over manner in which survey is completed (e.g. missing data, external assistance)<br>• Requires literacy (unless innovative audio-visual techniques used)<br>• If delivered online/via device, requires access to technology and user proficiency | Are instructions for completion clear? | If no, may result in response or non-response errors |
| | | | Are measures in place to minimise number of missed questions? | If no, may result in item non-response error |
| | | | Is the survey available in places that are convenient for / accessible to the target population? | If no, may result in sampling error |
| | | | Are there sufficient resources to ensure the survey is translated in languages required to obtain a representative of the target population? | If no, may result in sampling error |
| Interviewer-led | • Can facilitate rapport- and trust-building<br>• Enables greater control over the manner in which the survey is completed by participants Quality of responses is not dependent on participant literacy | • Lack of anonymity may increase social desirability response bias, especially when asking sensitive questions<br>• Requires participants to be present / available at the time the interviewers are able to collect data If delivered by telephone, requires eligible participants to have access Personnel costs May limit ability to collect data from places if interviewers need to travel long distances<br>• Limited to languages spoken by interviewers | Is the interview able to be offered at times that are convenient to the eligible population? | If no, may result in sampling error |
| | | | Are the interviewers appropriately trained? | If no, may result in interviewer error or processing error |
| | | | Are the characteristics of the available interviewers (e.g. gender) suitable given the characteristics of the participant and survey subject matter? | If no, may result in response error (social desirability), sampling error, interviewer error |
| | | | Are the interviews able to be conducted/offered in a private place? | If no, may result in response error (social desirability bias), sampling error |

with relevant guidelines (e.g. the *Guidelines for Best Practice in Cross-Cultural Surveys* [135]).

The main limitation of this scoping review is that it relied solely on information reported in the included studies. It is possible that a larger number of studies and data may have been included if authors had been approached to provide more information about the way in which their surveys were administered. Better reporting in the form of adherence to checklists such as STROBE [136] for observational studies and survey-specific guidelines (see Bennett, Khangura et al. [127]) will assist future researchers to undertake more comprehensive reviews into this subject area and facilitate their ability to produce rigorous meta-syntheses. Future reviews of survey research in migrant populations would also benefit from using appropriate tools to critically appraise the quality of included studies (e.g. checklists developed by the Joanna Briggs Institute or the Critical Appraisal Skills Program) [137, 138].

As Méndez and Font [139] note, "[t]he demand for more data about immigrants and ethnic minorities from national and supra-national bodies makes us confident that the number of surveys addressed to these populations will increase in the future." The challenge for

researchers is to ensure that future SHBBV surveys are designed with reference to, and with a view to building on, the evidence base about which mode of survey administration is best suited to collecting valid and reliable evidence about migrants' knowledge, behaviours and practices. Additionally, other factors influencing quality should also be examined, including sampling methods, survey translation and instrument validation. Research in this area is particularly salient, given the World Health Organizations current project to develop a "standard, globally-recognized instrument to measure sexual practices, behaviours and sexual health related outcomes" [15] which would facilitate comparisons across populations.

## Supporting information

**S1 Checklist. Preferred Reporting Items for Systematic reviews and Meta-Analyses extension for Scoping Reviews (PRISMA-ScR) checklist.**
(PDF)

**S1 Table. Search strategy for scoping review, by concept and database.**
(DOCX)

**S2 Table. Architecture for excel data charting table.**
(DOCX)

**S3 Table. Architecture for excel data charting table.**
(DOCX)

## Author Contributions

**Conceptualization:** Daniel Vujcich, Roanna Lobo, Bruce Maycock, Alison Reid.

**Data curation:** Daniel Vujcich, Sonam Wangda, Meagan Roberts, Chanaka Kulappu Thanthirige.

**Formal analysis:** Daniel Vujcich, Sonam Wangda.

**Funding acquisition:** Roanna Lobo, Bruce Maycock, Alison Reid.

**Methodology:** Daniel Vujcich, Sonam Wangda, Roanna Lobo, Bruce Maycock, Alison Reid.

**Project administration:** Daniel Vujcich, Meagan Roberts.

**Supervision:** Daniel Vujcich, Roanna Lobo, Bruce Maycock, Alison Reid.

**Validation:** Meagan Roberts, Roanna Lobo, Chanaka Kulappu Thanthirige.

**Writing – original draft:** Daniel Vujcich, Sonam Wangda.

**Writing – review & editing:** Daniel Vujcich, Sonam Wangda, Meagan Roberts, Roanna Lobo, Bruce Maycock, Chanaka Kulappu Thanthirige, Alison Reid.

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
