## [Decision Letter · Decision Letter 0]

13 May 2020

PONE-D-20-04548

Modes of administering sexual health and blood-borne virus surveys in migrant populations: a scoping review

PLOS ONE

Dear Dr Vujcich,

Thank you for submitting your manuscript to PLOS ONE. After careful consideration, we feel that it has merit but does not fully meet PLOS ONE’s publication criteria as it currently stands. Therefore, we invite you to submit a revised version of the manuscript that addresses the points raised during the review process.

We would appreciate receiving your revised manuscript by Jun 27 2020 11:59PM. To enhance the reproducibility of your results, we recommend that if applicable you deposit your laboratory protocols in protocols.io, where a protocol can be assigned its own identifier (DOI) such that it can be cited independently in the future. For instructions see: http://journals.plos.org/plosone/s/submission-guidelines#loc-laboratory-protocols

We look forward to receiving your revised manuscript.

Kind regards,

Chaisiri Angkurawaranon

Academic Editor

PLOS ONE

Journal Requirements:

Additional Editor Comments (if provided):

Reviewers' comments:

Reviewer's Responses to Questions

**Comments to the Author**

1. Is the manuscript technically sound, and do the data support the conclusions?

Reviewer #1: Yes

Reviewer #2: Partly

2. Has the statistical analysis been performed appropriately and rigorously? 

Reviewer #1: Yes

Reviewer #2: N/A

3. Have the authors made all data underlying the findings in their manuscript fully available?

Reviewer #1: Yes

Reviewer #2: Yes

4. Is the manuscript presented in an intelligible fashion and written in standard English?

Reviewer #1: Yes

Reviewer #2: Yes

5. Review Comments to the Author

Reviewer #1: PLoS ONE Manuscript #: PONE-D-20-04548

Title: Modes of administering sexual health and blood-borne virus surveys in migrant populations: a scoping review

Summary: This is a scoping review that explores how surveys focused on sexual health and blood-borne viruses have been developed and administered for migrant populations. In particular this study seeks to understand the manner in which such surveys are conducted among migrants and how this may relate to data quality, reliability, and bias represented in response rates and/or social desirability among participants.

This is a strong addition to the literature in my opinion. For the most part the article is well-prepared, well thought out, and well argued. The authors provide sound and strong recommendations.

I think this could very easily be accepted for publication with some minor revisions, as provided below. Most of my suggestions are for the introduction and discussion. I hope these and other suggestions might add to the strength of this manuscript.

Keywords:

Just a minor point of advice I find helpful, I always take my abstract and plug it into the “MESH on Demand” website, so that it can identify MESH keywords. I recommend this, as I think it will allow this important piece to be more easily found across web databases. When I do this for this article, I find keywords such as “sexual health,” “transients and migrants,” “social desirability”, “surveys and questionnaires”.

Abstract:

Line 31: It’s a bit unusual to see a citation in an abstract. I wonder if the authors may double-check and ensure that this is OK by PLoS ONE standards. I may state the methodological framework as done in the methods, “methodological framework for scoping reviews”—the authors may get around citation that way.

Introduction:

I think the introduction needs a bit of work to tighten up the language and for logical flow. This may be accomplished by rewording to a more active voice, better use of transition phrases to help better guide the reader, and re-ordering some of the clauses/statements.

Line 55: I think the first sentence can be written in a more active voice. Consider “UNAIDS prioritizes migrants as an at-risk group that requires a response…”

Lines 58-60: I may rephrase this to flow better and make the authors’ argument more concise and a bit “punchier”: “Elsewhere such as in the USA and Australia, migrants accounted for XX% and XX% of HIV diagnoses respectively.”

Line 61-62: “barriers to health-seeking behaviours” – can one, especially migrants, actually have barriers to health-seeking behaviours? Or is it more appropriate to say barriers to health access or to being evaluated/screen for SHBBV?

Line 64: can delete “sources”

Lines 64-67: I would switch the first two sentences in the second paragraph. I would begin with, “In spite of the priority for this vulnerable population, migrants are still under-represented in research…”. Then, follow with “High quality data are required to address SHBBV in migrants.” I think this would flow better into the third sentence of the paragraph.

Line 68: “administered to” can be changed to “developed for”—it’s a bit more inclusive than just administering, it suggests that these surveys are actually designed with migrants in mind, more central to your study.

I think paragraphs 3 and 4 need to be combined, with the argument in paragraph 4 placed earlier.

I would start paragraph 3 with a statement more like, “The mode by which a survey is administered can greatly affect the quality of the data collected.” This should be followed by the Italian study—but can this be cut down from two sentences to one?

Then follow with, “As a recent literature review shows, the manner of survey administration can greatly affect the quality of the data collected by influencing response rates…”. Then close with “However, this review did not seek…”

The 5th paragraph can start with a different introductory sentence, for example: “How these differing modes of survey administration affect data quality can be even more complicated among migrants.” Then follow with “For example, it has been noted…”. Then flip the following two sentences. The last sentence should then read, “Likely reflective of such concerns, a recent review of 550 surveys…”.

Begin the 6th paragraph with, “Although this apparent tension between sensitivity and accessibility can effect data quality, there is still no strong/empirical guidance to determine appropriate modes of SHBBV survey administration among migrant populations.” Or some such.

I actually would combine the last paragraph with the 6th paragraph. After the Font and Mendez quote, follow with, “Therefore, we aimed to perform a scoping review of SHBBV surveys administered to migrant populations, understand the effect that mode of administration as on key indicators of survey quality, such as data reliability, response rates, and social desirability bias.”

Materials and methods:

Not only are the methods strong, they are well explained. Well done.

Line 111: Some other reviewers may give the authors trouble on this, so I would consider rephrasing here. Start with what is already written, “The broad research objective….” The following sentence could state, “Hence, the objective could be answered through the following questions, which were used to determine studies to be included in our scoping review”. Some reviewers may struggle that these questions weren’t informed by the literature. But if asked, I recommend that the authors might cite some of the studies from the introduction that prompted these particular questions (such as the literature review).

Line 119-120 “This scoping review was conducted as a sub-study…” – I think this sentence can be omitted.

Line 124: For tables and figures, I wouldn’t write where they occur in relation to the text, because this might change over the course of the publication.

Table 1: I think this can be a supplementary table.

Results: Another strong section.

Line 157, 158-159: STI and BBV – why change the acronym here? Did the authors mean to keep to SHBBV?

Table 3 and some of the figures as well: Double-check because LAC isn’t always completely spelled out as an acronym. Helpful to clarify it in the figures and tables as well as in the text (as has been done).

Line 166-172: consider some editing here for clarity, brevity, and flow. Try to get the 4 sentences into 2 or 3 max.

Line 183: Give the work some credit here by adding a clause, “Given the level of reporting by the studies included, it was not possible…”.

Discussion:

I think the second paragraph makes for a stronger, “introductory” paragraph to the discussion. I would put this first, however, I may eschew this quote as I don’t fully understand it’s significance. Change “glean” for “determine” – the methodology was much more thorough than just “gleaning!” I would place paragraph 2 before the first paragraph of the Discussion. If re-ordered this way, the claim of “lack of transparency” (line 279) in paragraph 3 makes more sense.

Paragraph 3 is great, but it’s not reflected in the introduction. As a researcher on migrant health, I felt this was part of the impetus for this study as well—the ethical considerations. Is it possible to briefly point this out in the introduction when the authors discuss that modes of delivery are more complicated among migrants?

Line 296: Does this “reveal a debate” as much as “broach some important considerations” or some such?

Line 304-305: Should it be “Poynton, DeFouw, et al.”?

Table 5: We may have to defer to the editor, but I would rather see this in the results. I would possibly link this with the numbered list as explanation as Table 5 doesn’t have citations (though you may include important ones in the footnotes of the table). However, I consider all of this to come from the authors’ analysis of the studies in their review.

Line 352-353: the authors mentioned including grey literature studies in the methods, but over the course of the results and here in the Discussion, it seems that no studies in the grey literature were included in the analysis? Would this be best clarified in the Results?

References:

Double-check so that these are edited appropriately, according to PLoS ONE guidelines. I note missing punctuation, capitalization different for some entries, years not reported consistently (in terms of placement), etc.

Reviewer #2: This paper was interesting to me as a migration and health researcher. We have debated the ethics of asking migrants about sexual health, in particular the ethical challenges of taking blood samples for diagnoses, particularly in countries where BBV are deportable conditions (e.g. Malaysia, Singapore).

My main concern with this paper is in the reporting. The authors should be aware that there is a PRISMA extension for Scoping reviews.

PRISMA-ScR checklist: http://www.prisma-statement.org/Extensions/ScopingReviews

2018 Tricco elaboration article: https://annals.org/aim/fullarticle/2700389/prisma-extension-scoping-reviews-prisma-scr-checklist-explanation

Using an earlier methodological paper defining scoping reviews from Arksey and O’Malley is a limitation. As the PRISMA-ScR was not followed, I suggest that the authors state this as a Limitation in the Discussion section. Alternatively, the authors can consider re-writing the paper to follow PRISMA guidelines. The foundation and steps followed are in the manuscript, but it does require some rewriting and expansion of the Methods section, to include details like were data extraction forms piloted and by whom, etc.

My second comment that applies to the overall manuscript, is that the authors sufficiently caveat, in the Introduction and Discussion, that mode of survey administration is one of numerous factors affecting data quality. Specific suggestions are offered below.

Methods:

l. 107 – please insert a brief explanation describing what domains the Arksey and O’Malley framework covers. Here, authors should explicitly state that the PRISMA-ScR was not followed and the reason for this. The authors can consider including a figure depicting the Arksey and O’Malley domains for ease of reading.

l. 119 – after describing the objectives, the inclusion/exclusion criteria (and correspondingly Table 2) should be placed before information on searches and search strings. This is the preferred order in PRISMA reviews, with the reason that we need to see the eligibility criteria before assessing search terms etc against them. I’d suggest including some brief narrative text on inclusion/exclusion criteria alongside Table 2.

l. 136 – Bramer and Giustini framework – suggest to add ‘for this purpose’ at end of the sentence. Generally when introducing a framework, a brief explanation of its content will help readers who are not familiar with these articles.

Discussion:

l. 268 – Citing study specific reporting guidelines would be helpful after ref. 113, e.g. STROBE for observational studies. List here: https://www.equator-network.org/

l. 273 – on conducting surveys more efficiently and effectively to produce reliable data - Mode of administration is just one factor affecting response rates. Other factors include how survey constructs were developed and validated in local settings, whether it was piloted and cultural/linguistic adjustments made, inter-rater reliability (e.g. enumerator training to ensure standardized answers across enumerators) and sufficient ethical procedures to ensure participants that data collected is confidential, anoynmized, etc. For patient reported outcome measures (PROMS), there is something called the COSMIN standards which include a 114-item checklist to assess content validity of outcome measures, with a section on reliability. Line 288 goes on to ask whether methods produced valid and reliable data. To assess this comprehensively, we need to conduct critical appraisal at the level of the individual study (e.g using. JBI, CASP tools) and at the level of the outcome measure (e.g. COSMIN tool, or shorter appraisal checklists which examine outcome measures only, e.g. 5 item checklist in this article). I’d suggest the authors refer to critical appraisal tools in this section, to offer specific guidance to researchers looking to improve the reliability and validity of survey data they collect with migrant populations.

Whether the included studies used validated SRH or SHBBV survey modules, vs. researchers making up their own questions, is not specified. This is especially important given the sensitive nature of the research topic. Using validated measures can help generate reliable data. The authors should mention somewhere in the Discussion, the rough proportion of studies which used validated (or at least, established) measures, and what the most commonly used measures are called/which larger surveys they were taken from (e.g. NATSAL in the UK/DHS globally. Other example measures on p.4 in this document). Were any of the measures validated with migrant populations as well?

Searching online, I was shocked to see there is no internationally validated or recommended instrument (https://www.who.int/docs/default-source/reproductive-health/sexual-health-survey-instrument-info-faq.pdf?sfvrsn=3172d357_2) – I’d suggest that the authors mention that the WHO is actively soliciting submissions from researchers to recommend appropriate survey modules, in the Discussion.

l. 345 recommendation 4 – suggest to include that authors specify that culturally appropriate adjustments are made (where needed) after piloting surveys.

l. 355 – reference to better reporting – suggest to cite STROBE, CONSORT etc specifically to help guide researchers in the right direction.

l. 352 – the Limitation section should be expanded with the main limitation that PRISMA-ScR guidelines were not followed. While not required for scoping reviews, the lack of critical appraisal (at the level of the individual study, and outcome measure) can be considered a limitation, given that reliability of data will be affected by study design decisions, and content validity of constructs in survey modules.

PRISMA flow diagram – if available, authors should insert specific reasons for exclusion at the full-text stage.

6. PLOS authors have the option to publish the peer review history of their article (what does this mean?). If published, this will include your full peer review and any attached files.

Reviewer #1: No

Reviewer #2: Yes: Nicola Pocock

---

## [Author Response · Author response to Decision Letter 0]

26 Jun 2020

Dear Reviewers, 

PONE-D-20-04548: RESPONSE TO REVIEWERS

Thank you for your considered and helpful feedback on our manuscript titled Modes of administering sexual health and blood-borne virus surveys in migrant populations: a scoping review. 

We have reproduced and responded to each item of feedback below. Please note that the line numbers that we cite refer to the marked-up version of the manuscript. 

1. ABSTRACT

Reviewer 1: Line 31: It’s a bit unusual to see a citation in an abstract. I wonder if the authors may double-check and ensure that this is OK by PLoS ONE standards. I may state the methodological framework as done in the methods, “methodological framework for scoping reviews”—the authors may get around citation that way.

Response:

We have removed the Arksey and O’Malley citation and reworded the sentence to state: “A methodological framework for scoping reviews was applied” (lines 31-32). 

2. KEYWORDS 

Reviewer 1: I always take my abstract and plug it into the “MESH on Demand” website, so that it can identify MESH keywords. I recommend this, as I think it will allow this important piece to be more easily found across web databases. When I do this for this article, I find keywords such as “sexual health,” “transients and migrants,” “social desirability”, “surveys and questionnaires”. 

Response: 

We have followed your useful advice. The keywords are now: surveys and questionnaires; sexual health; migrants; data accuracy; bias. 

3. GENERAL ISSUES 

3.1. Reporting style

Reviewer 2: My main concern with this paper is in the reporting. The authors should be aware that there is a PRISMA extension for Scoping reviews … As the PRISMA-ScR was not followed, I suggest that the authors state this as a Limitation in the Discussion section. Alternatively, the authors can consider re-writing the paper to follow PRISMA guidelines. The foundation and steps followed are in the manuscript, but it does require some rewriting and expansion of the Methods section, to include details like were data extraction forms piloted and by whom, etc.

Response: 

Thank you for bringing this to our attention. We have rewritten the paper to follow the PRISMA-ScR guidelines. A completed PRISMA-ScR checklist is attached for your reference. Please note that the page numbers referred to in the checklist relate to the untracked version of the manuscript. 

3.2. Inclusion of caveats

Reviewer 2: My second comment that applies to the overall manuscript, is that the authors sufficiently caveat, in the Introduction and Discussion, that mode of survey administration is one of numerous factors affecting data quality. Specific suggestions are offered below.

Response: 

The following sentence has been added to the Introduction: “While there are a range of factors which can affect the quality of survey data (e.g. validity of survey constructs, sampling and recruitment methods), the focus of this article is the mode of survey administration” (lines 83-85). 

The underlined sentence has been added to the Discussion: “The challenge for researchers is to ensure that future SHBBV surveys are designed with reference to, and with a view to building on, the evidence base about which mode of survey administration is best suited to collecting valid and reliable evidence about migrants’ knowledge, behaviours and practices. Additionally, other factors influencing quality should also be examined, including sampling methods, survey translation and instrument validation” (lines 382-384). 

The review’s specific suggestions have also been followed, as detailed below. 

4. INTRODUCTION 

Reviewer 1: I think the introduction needs a bit of work to tighten up the language and for logical flow. This may be accomplished by rewording to a more active voice, better use of transition phrases to help better guide the reader, and re-ordering some of the clauses/statements.

Response: 

The Introduction has been amended in line with your specific suggestions below. 

Reviewer 1 suggestions 

Line 55: I think the first sentence can be written in a more active voice. Consider “UNAIDS prioritizes migrants as an at-risk group that requires a response…”

Response: 

Changed to: “Migrants are a priority group in the prevention and control of HIV/AIDS” (line 56). 

Reviewer 1 suggestions 

Line 58-60: I may rephrase this to flow better and make the authors’ argument more concise and a bit “punchier”: “Elsewhere such as in the USA and Australia, migrants accounted for XX% and XX% of HIV diagnoses respectively.” 

Response: 

Amended as suggested (line 61). 

Reviewer 1 suggestions 

Line 61-62: “barriers to health-seeking behaviours” – can one, especially migrants, actually have barriers to health-seeking behaviours? Or is it more appropriate to say barriers to health access or to being evaluated/screen for SHBBV? 

Response: 

Replaced with “health care access” (line 64). 

Reviewer 1 suggestions 

Line 64: can delete “sources” 

Response: 

Amended as suggested (line 68).

Reviewer 1 suggestions 

Lines 64-67: I would switch the first two sentences in the second paragraph. I would begin with, “In spite of the priority for this vulnerable population, migrants are still under-represented in research…”. Then, follow with “High quality data are required to address SHBBV in migrants.” I think this would flow better into the third sentence of the paragraph. 

Response: 

Amended as suggested (lines 66-68). 

Reviewer 1 suggestions 

Line 68: “administered to” can be changed to “developed for”—it’s a bit more inclusive than just administering, it suggests that these surveys are actually designed with migrants in mind, more central to your study. 

Response: 

Amended as suggested (lines 72-73).

 Reviewer 1 suggestions 

I think paragraphs 3 and 4 need to be combined, with the argument in paragraph 4 placed earlier. I would start paragraph 3 with a statement more like, “The mode by which a survey is administered can greatly affect the quality of the data collected.” This should be followed by the Italian study—but can this be cut down from two sentences to one? Then follow with, “As a recent literature review shows, the manner of survey administration can greatly affect the quality of the data collected by influencing response rates…”. Then close with “However, this review did not seek…” 

Response: 

We have combined paragraphs 3 and 4 (lines 83 to 96). We have not changed the current order in which the studies are referred. We feel that it is first important to introduce the literature about survey mode generally, before referring to the literature that is specific to sexual health studies. The summary of the Italian study has been shortened to one sentence as suggested. 

Reviewer 1 suggestions 

The 5th paragraph can start with a different introductory sentence, for example: “How these differing modes of survey administration affect data quality can be even more complicated among migrants.” Then follow with “For example, it has been noted…”. Then flip the following two sentences. The last sentence should then read, “Likely reflective of such concerns, a recent review of 550 surveys…”.

Response: 

Amended as suggested (line 104 onwards). 

Reviewer 1 suggestions 

Begin the 6th paragraph with, “Although this apparent tension between sensitivity and accessibility can effect data quality, there is still no strong/empirical guidance to determine appropriate modes of SHBBV survey administration among migrant populations.” Or some such.

I actually would combine the last paragraph with the 6th paragraph. After the Font and Mendez quote, follow with, “Therefore, we aimed to perform a scoping review of SHBBV surveys administered to migrant populations, understand the effect that mode of administration as on key indicators of survey quality, such as data reliability, response rates, and social desirability bias.” 

Response: 

The paragraph now begins with a statement about the ethical obligations around the collection of sensitive data from vulnerable populations (in response to your feedback in the row below) (line 119 onwards). The paragraph then proceeds as suggested. We have combined paragraphs 5 and 6. 

Reviewer 1 suggestions:

Paragraph 3 [of the Discussion] is great, but it’s not reflected in the introduction. As a researcher on migrant health, I felt this was part of the impetus for this study as well—the ethical considerations. Is it possible to briefly point this out in the introduction when the authors discuss that modes of delivery are more complicated among migrants? 

Response: 

The following sentence has been added to the introduction: “When collecting sensitive data from potentially vulnerable populations, researchers have an ethical imperative to ensure that any foreseeable harms are proportionate to the benefits that can flow from valid and reliable research outputs” (lines 119-121). 

5. MATERIALS AND METHODS 

5.1. Explanation of Arksey and O’Malley framework 

Reviewer 2: Please insert a brief explanation describing what domains the Arksey and O’Malley framework covers. Here, authors should explicitly state that the PRISMA-ScR was not followed and the reason for this. The authors can consider including a figure depicting the Arksey and O’Malley domains for ease of reading. 

Response: 

A summary of the Arksey and O’Malley domains is now included as Table 1. 

5.2. Statement of objectives 

Reviewer 1: Line 111: Some other reviewers may give the authors trouble on this, so I would consider rephrasing here. Start with what is already written, “The broad research objective….” The following sentence could state, “Hence, the objective could be answered through the following questions, which were used to determine studies to be included in our scoping review”. Some reviewers may struggle that these questions weren’t informed by the literature. But if asked, I recommend that the authors might cite some of the studies from the introduction that prompted these particular questions (such as the literature review).

Response: 

Amended as follows: “The broad research objective was to determine what modes of survey administration have been used to conduct sexual health and blood-borne virus surveys in migrant populations and to ascertain the strengths and limitations associated with each mode. The objective was complemented by following sub-questions were set to meet the stated objective …” (lines 142-143).

5.3. Order of inclusion and exclusion criteria 

Reviewer 2: After describing the objectives, the inclusion/exclusion criteria (and correspondingly Table 2) should be placed before information on searches and search strings. This is the preferred order in PRISMA reviews, with the reason that we need to see the eligibility criteria before assessing search terms etc against them. I’d suggest including some brief narrative text on inclusion/exclusion criteria alongside Table 2.

Response: 

Amended as suggested. Please see lines 152 onwards. 

5.4. Search strategy table 

Reviewer 1: Table 1 - I think this can be a supplementary table.

Response: 

Amended as suggested. 

5.5. Explanation of Bramer and Guistini framework 

Reviewer 2: Bramer and Giustini framework – suggest to add ‘for this purpose’ at end of the sentence. Generally when introducing a framework, a brief explanation of its content will help readers who are not familiar with these articles.

Response: 

The underlined text has been added to this sentence: “Results were imported into Endnote and de-duplicated using the process developed by Bramer, Giustini (21) for this purpose (e.g. tailored use of field settings and filters)” (line 187). 

5.6. Other suggestions for this section 

Reviewer 1 suggestions 

“This scoping review was conducted as a sub-study…” – I think this sentence can be omitted.

Response: 

Amended as suggested (line 163).

Reviewer 1 suggestions 

For tables and figures, I wouldn’t write where they occur in relation to the text, because this might change over the course of the publication.

Response: 

Amended as suggested

6. RESULTS 

6.1. PRISMA flow diagram 

Reviewer 2: If available, authors should insert specific reasons for exclusion at the full-text stage.

Response: 

Amended as suggested – please see revised Figure 1. 

Please note that in the course of reviewing data on reasons for exclusion, we identified three studies which ought to have been included in the results but were inadvertently omitted from the first submitted draft. These studies have now been included in the revised manuscript and the findings have been amended accordingly. The inclusion of these studies does not affect the overall results or conclusions. 

6.2. Acronyms 

Reviewer 1: 

• Line 157, 158-159: STI and BBV – why change the acronym here? Did the authors mean to keep to SHBBV?

• Table 3 and some of the figures as well: Double-check because LAC isn’t always completely spelled out as an acronym. Helpful to clarify it in the figures and tables as well as in the text (as has been done).

Response: 

We have added a note on LAC abbreviation in Table 3 and we have removed the abbreviation from Figure 3. 

‘STI and BBV’ has been changed to ‘SHBBV’ throughout.

6.3. Editing for clarity and brevity 

Reviewer 1: Lines 166-172 - consider some editing here for clarity, brevity, and flow. Try to get the 4 sentences into 2 or 3 max.

Response: 

Amended as suggested to now read: “As shown in Figure 2, ‘interview only’ was the most common mode of survey administration (n=48), with face-to-face (n=37) being the most common interview technique. Of the thirty three studies reporting data from ‘self-completed’ surveys only pen-and-paper was the most common method of self-completion (n=15). Few studies (n=7) combined interview and self-completed methods of survey administration” (lines 246-253). 

6.4. Addition of a clause 

Reviewer 1: Lines 183 - Give the work some credit here by adding a clause, “Given the level of reporting by the studies included, it was not possible…”.

Response: 

Amended to now read: “Given the level of reporting in the included studies, it was not possible to determine whether certain modes of administration were associated with higher response rates, controlling for other factors” (lines 264-266). 

7. DISCUSSION 

7.1. Changes to paragraphs one and two 

Reviewer 1: I think the second paragraph makes for a stronger, “introductory” paragraph to the discussion. I would put this first, however, I may eschew this quote as I don’t fully understand it’s significance. Change “glean” for “determine” – the methodology was much more thorough than just “gleaning!” I would place paragraph 2 before the first paragraph of the Discussion. If re-ordered this way, the claim of “lack of transparency” (line 279) in paragraph 3 makes more sense.

Response: 

Amended as suggested. 

7.2. Cite specific study reporting guidelines

Reviewer 2: Line 268 - Citing study specific reporting guidelines would be helpful after ref. 113, e.g. STROBE for observational studies. List here: https://www.equator-network.org/

Response: 

Amended as follows: “These findings emphasise the need for greater adherence to (or awareness of) reporting standards. For instance, the STROBE checklist for observational studies requires details about setting (e.g. recruitment sites and sources), eligibility criteria, method of recruitment, and numbers of individuals at each stage of the study, and reasons for non-participation at each stage (126). Survey-specific checklists also recommend reporting additional details including description of the survey instrument and its development, pretesting processes, instrument reliability and validity, sample representativeness, mode of administration, number of attempts made to contact subjects, whether incentives were offered, methods for analysis of nonresponse error and descriptions of consent procedures (see data extraction tool published by Bennett, Khangura et al. (127))” (lines 273-286). 

7.3. Refer to critical appraisal tools

Reviewer 2: On conducting surveys more efficiently and effectively to produce reliable data - Mode of administration is just one factor affecting response rates. Other factors include how survey constructs were developed and validated in local settings, whether it was piloted and cultural/linguistic adjustments made, inter-rater reliability (e.g. enumerator training to ensure standardized answers across enumerators) and sufficient ethical procedures to ensure participants that data collected is confidential, anoynmized, etc. For patient reported outcome measures (PROMS), there is something called the COSMIN standards which include a 114-item checklist to assess content validity of outcome measures, with a section on reliability. Line 288 goes on to ask whether methods produced valid and reliable data. To assess this comprehensively, we need to conduct critical appraisal at the level of the individual study (e.g using. JBI, CASP tools) and at the level of the outcome measure (e.g. COSMIN tool, or shorter appraisal checklists which examine outcome measures only, e.g. 5 item checklist in this article). I’d suggest the authors refer to critical appraisal tools in this section, to offer specific guidance to researchers looking to improve the reliability and validity of survey data they collect with migrant populations.

Response: 

Critical appraisal tools are now referred to in the Discussion: “Better reporting in the form of adherence to checklists such as STROBE (136) for observational studies and survey-specific guidelines (see Bennett, Khangura et al. (127)) will assist future researchers to undertake more comprehensive reviews into this subject area and facilitate their ability to produce rigorous meta-syntheses. Future reviews of survey research in migrant populations would also benefit from using appropriate tools to critically appraise the quality of included studies (e.g. checklists developed by the Joanna Briggs Institute or the Critical Appraisal Skills Program) (137, 138)” (lines 368-375).

7.4. Information on survey validation 

Reviewer 2: Whether the included studies used validated SRH or SHBBV survey modules, vs. researchers making up their own questions, is not specified. This is especially important given the sensitive nature of the research topic. Using validated measures can help generate reliable data. The authors should mention somewhere in the Discussion, the rough proportion of studies which used validated (or at least, established) measures, and what the most commonly used measures are called/which larger surveys they were taken from (e.g. NATSAL in the UK/DHS globally. Other example measures on p.4 in this document). Were any of the measures validated with migrant populations as well?

Response: 

We have included this information in the Results. Table 3 has been amended to include a new column labelled “SHBBV instrument used”. The Table is followed by the following paragraph: “In 44 studies, existing SHBBV instruments were adapted or used. These instruments included the Brief HIV Screener (116), the Perceived Susceptibility to HIV Scale (117), the AIDS Health Belief Scale (118), the National Survey of Australian Secondary Students and Sexual Health (119), the Survey of Latino Adults (120, 121), the African Health and Sex Survey (13), the Bass Line Survey (122), UN Behavioral Surveillance Surveys (123) and the National Health Interview Survey Supplement on AIDS Knowledge and Attitudes (124). For the remaining studies, either the origin of the survey items was not reported (n=23) or new survey instruments were developed (n=24). Of the studies in which new survey instruments were developed, half (n=12) did not explicitly report whether the instrument had been pretested or piloted” (lines 235-244). 

7.5. Reference to WHO project to develop a standardised SHBBV instrument 

Reviewer 2: Searching online, I was shocked to see there is no internationally validated or recommended instrument (https://www.who.int/docs/default-source/reproductive-health/sexual-health-survey-instrument-info-faq.pdf?sfvrsn=3172d357_2) – I’d suggest that the authors mention that the WHO is actively soliciting submissions from researchers to recommend appropriate survey modules, in the Discussion.

Response: 

The following sentence has been added: “Research in this area is particularly salient, given the World Health Organizations current project to develop a “standard, globally-recognized instrument to measure sexual practices, behaviours and sexual health related outcomes” (15) which would facilitate comparisons across populations” (lines 384-387). The WHO initiative is also now mentioned in the Introduction (lines 79-82).

7.6. Addition to Recommendation 4 

Reviewer 2: Recommendation 4 – suggest to include that authors specify that culturally appropriate adjustments are made (where needed) after piloting surveys.

Response: 

The following sentence has been added: “Where issues are identified through pre-testing and piloting, appropriate revisions should be made in line with relevant guidelines (e.g. the Guidelines for Best Practice in Cross-Cultural Surveys [135])” (lines 362-364). 

7.7. Position of Table 5

Reviewer 1: We may have to defer to the editor, but I would rather see this in the results. I would possibly link this with the numbered list as explanation as Table 5 doesn’t have citations (though you may include important ones in the footnotes of the table). However, I consider all of this to come from the authors’ analysis of the studies in their review.

Response: 

In our view, Table 5 sets out the broader significance of the results presented after Table 4. For that reason we have left it in the discussion section but we will defer to the Editor’s judgment.

7.8. Reference to grey literature 

Reviewer 1: Lines 352-353: the authors mentioned including grey literature studies in the methods, but over the course of the results and here in the Discussion, it seems that no studies in the grey literature were included in the analysis? Would this be best clarified in the Results?

Response: 

The following items in Table 3 are grey literature sources: Ahmed (2013), Amadi (2012), Chamratrithirong etal (2005), Cohen (2015), Demeke (2013), Johnston (2019), Kara (2012), Ogungbade (2010), UNHCR (2004, 2006).

7.9. Limitations section 

Reviewer 2: 

• The Limitation section should be expanded with the main limitation that PRISMA-ScR guidelines were not followed. While not required for scoping reviews, the lack of critical appraisal (at the level of the individual study, and outcome measure) can be considered a limitation, given that reliability of data will be affected by study design decisions, and content validity of constructs in survey modules.

• Line 355 - reference to better reporting – suggest to cite STROBE, CONSORT etc specifically to help guide researchers in the right direction.

Response:

The manuscript has been amended to be consistent with PRISMA-ScR guidelines.

The following sentence has also been added: “Future reviews of survey research in migrant populations would also benefit from using appropriate tools to critically appraise the quality of included studies (e.g. checklists developed by the Joanna Briggs Institute or the Critical Appraisal Skills Program) (137, 138)” (lines 372-375).

The sentence about ‘better reporting’ has also be amended as follows: “Better reporting in the form of adherence to checklists such as STROBE (136) for observational studies and survey-specific guidelines (see Bennett, Khangura et al (127)) will assist future researchers to undertake more comprehensive reviews into this subject area and facilitate their ability to produce rigorous meta-syntheses” (lines 368-370).

7.10. Other suggestions for this section 

Reviewer 1 suggestions 

Line 296: Does this “reveal a debate” as much as “broach some important considerations” or some such?

Response:

Amended as suggested

Reviewer 1 suggestions 

Line 304-305: Should it be “Poynton, DeFouw, et al.”? 

Response:

Amended as suggested

8. REFERENCES

Reviewer 1: Double-check so that these are edited appropriately, according to PLoS ONE guidelines. I note missing punctuation, capitalization different for some entries, years not reported consistently (in terms of placement), etc.

Response:

Edits have been made to the reference lists as suggested. 

We are grateful for the contributions you have made to help improve this paper. We hope that our revisions have adequately addressed the issues you have identified. 

Yours sincerely, 

The authors

17 June 2020

---

## [Decision Letter · Decision Letter 1]

15 Jul 2020

Modes of administering sexual health and blood-borne virus surveys in migrant populations: a scoping review

PONE-D-20-04548R1

Dear Dr. Vujcich,

We’re pleased to inform you that your manuscript has been judged scientifically suitable for publication and will be formally accepted for publication once it meets all outstanding technical requirements.

Kind regards,

Chaisiri Angkurawaranon

Academic Editor

PLOS ONE

Additional Editor Comments (optional):

Reviewers' comments:

Reviewer's Responses to Questions

**Comments to the Author**

1. If the authors have adequately addressed your comments raised in a previous round of review and you feel that this manuscript is now acceptable for publication, you may indicate that here to bypass the “Comments to the Author” section, enter your conflict of interest statement in the “Confidential to Editor” section, and submit your "Accept" recommendation.

Reviewer #2: All comments have been addressed

2. Is the manuscript technically sound, and do the data support the conclusions?

Reviewer #2: Yes

3. Has the statistical analysis been performed appropriately and rigorously? 

Reviewer #2: N/A

4. Have the authors made all data underlying the findings in their manuscript fully available?

Reviewer #2: Yes

5. Is the manuscript presented in an intelligible fashion and written in standard English?

Reviewer #2: Yes

6. Review Comments to the Author

Reviewer #2: Thank you to the authors for revising this manuscript. I was really interested to see the proportion of studies using existing instruments vs. conceiving their own. I did not find the PRISMA Sc-R checklist with the resubmission - this should be included as supplementary information with the article.

7. PLOS authors have the option to publish the peer review history of their article (what does this mean?). If published, this will include your full peer review and any attached files.

Reviewer #2: **Yes: **Nicola Pocock

---

## [Editor Report · Acceptance letter]

23 Jul 2020

PONE-D-20-04548R1 

Modes of administering sexual health and blood-borne virus surveys in migrant populations: a scoping review 

Dear Dr. Vujcich:

I'm pleased to inform you that your manuscript has been deemed suitable for publication in PLOS ONE. Congratulations! Your manuscript is now with our production department. 

Kind regards, 

on behalf of

Dr. Chaisiri Angkurawaranon 

Academic Editor

PLOS ONE